# Enhancing Cervical Cancer Screening: Review of p16/Ki-67 Dual Staining as a Promising Triage Strategy

**DOI:** 10.3390/diagnostics14040451

**Published:** 2024-02-19

**Authors:** Yung-Taek Ouh, Ho Yeon Kim, Kyong Wook Yi, Nak-Woo Lee, Hai-Joong Kim, Kyung-Jin Min

**Affiliations:** Department of Obstetrics and Gynecology, Korea University Ansan Hospital, Ansan-si 15355, Gyeonggi-do, Republic of Korea; oytjjang@gmail.com (Y.-T.O.); shinbi7873@naver.com (H.Y.K.); kwyi@korea.ac.kr (K.W.Y.); nwlee@korea.ac.kr (N.-W.L.); haijkim@korea.ac.kr (H.-J.K.)

**Keywords:** cervical cancer, high-risk human papillomavirus, cervical intraepithelial neoplasia, Papanicolaou test, p16/Ki-67 dual staining, cytology, HPV testing

## Abstract

Cervical cancer, primarily caused by high-risk human papillomavirus (HR-HPV) types 16 and 18, is a major global health concern. Persistent HR-HPV infection can progress from reversible precancerous lesions to invasive cervical cancer, which is driven by the oncogenic activity of human papillomavirus (HPV) genes, particularly E6 and E7. Traditional screening methods, including cytology and HPV testing, have limited sensitivity and specificity. This review explores the application of p16/Ki-67 dual-staining cytology for cervical cancer screening. This advanced immunocytochemical method allows for simultaneously detecting p16 and Ki-67 proteins within cervical epithelial cells, offering a more specific approach for triaging HPV-positive women. Dual staining and traditional methods are compared, demonstrating their high sensitivity and negative predictive value but low specificity. The increased sensitivity of dual staining results in higher detection rates of CIN2+ lesions, which is crucial for preventing cervical cancer progression. However, its low specificity may lead to increased false-positive results and unnecessary biopsies. The implications of integrating dual staining into contemporary screening strategies, particularly considering the evolving landscape of HPV vaccination and changes in HPV genotype prevalence, are also discussed. New guidelines and further research are necessary to elucidate the long-term effects of integrating dual staining into screening protocols.

## 1. Introduction

Cervical cancer ranks as the fourth most prevalent malignancy among women globally and is primarily attributed to infection with high-risk human papillomavirus (HR-HPV), notably types 16 and 18 [1,2,3,4,5,6,7,8]. Persistent HR-HPV infection can progress from reversible precancerous lesions to invasive cervical cancer, driven by the actions of HPV genes, particularly E6 and E7, which lead to cell cycle dysregulation. Early detection through screening is pivotal for the prevention and management of cervical cancer [9]. Currently, cytology, HPV testing, and their combination are used for cervical cancer screening [10]. The Papanicolaou (Pap) test based on cytology has been instrumental in reducing the morbidity and mortality associated with cervical cancer [10]. However, cytology has limitations in terms of sensitivity and subjectivity, prompting the integration of HPV testing to enhance screening effectiveness [11,12].

Patients with mild cervical lesions, such as low-grade squamous intraepithelial lesions (LSIL) and atypical squamous cells of undetermined significance (ASCUS), face the risk of progression to more severe conditions, necessitating careful triage strategies [13]. Although HPV testing has high sensitivity, its low specificity leads to a significant number of unnecessary colposcopies, particularly in younger women [14]. In this context, p16/Ki-67 dual-staining cytology emerges as a potential biomarker with high sensitivity and specificity for identifying high-grade cervical intraepithelial neoplasia (HGCIN) [15,16].

Although HPV testing is sensitive, it faces challenges in terms of specificity, particularly in younger populations, making it less cost-effective as a separate screening tool for women aged <30 years [17,18]. Current screening strategies involve co-testing with HPV genotyping and cytology, leading to referral for colposcopy based on abnormal findings [19]. However, effective triage tests are needed to reduce unnecessary colposcopy and enhance the detection of CIN in HPV-positive women [20]. p16/Ki-67 dual-stained cytology has emerged as a promising biomarker, offering morphology-independent insights into cell cycle deregulation associated with HPV infections [21].

Although HPV testing has been a valuable addition to cervical cancer screening programs, enhancing early detection of precancerous lesions, its limitations underscore the need for additional triage strategies to optimize patient care and resource utilization [22]. This need highlights the evolving role of p16/Ki-67 dual-staining cytology, which promises to address some of these limitations by offering a more specific method for triaging HPV-positive women (Table 1).

## 2. P16/Ki-67 Dual Staining: An Emerging Tool for Triage

The p16/Ki-67 dual-staining technique is an advanced immunocytochemical method employed for cervical cancer screening [23]. This technique allows the simultaneous detection of p16 and Ki-67 proteins within the same cervical epithelial cell [24].

The utilization of p16INK4a in cervical cancer screening presents a promising alternative to existing triage strategies for women with abnormal Papanicolaou (Pap) results [25,26]. Current approaches, including repeat cytology, HPV testing, and colposcopy-guided biopsy, face challenges such as high referral rates and limited specificity [27]. P16INK4a, induced by HR-HPV oncogenes, exhibits a more specific association with HGCIN [28,29]. A qualitative analysis of cytological specimens, focusing on nuclear alterations, enhances specificity and enables the detection of the underlying HGCIN [30]. Studies have demonstrated that p16INK4acytology outperforms HPV testing in triaging patients with ASCUS and LSIL, demonstrating increased specificity (from approximately 50% to 80%) [26]. The p16INK4a-based approach exhibits high sensitivity and specificity for identifying HGCIN, suggesting its potential as a reliable and reproducible triage [31,32]. Its use could reduce the number of colposcopy referrals and enhance automated preselection of suspicious slides, offering an efficient and accurate tool in cervical cancer screening [33,34,35].

p16 is a dependent kinase inhibitor primarily involved in controlling the transition from the G1 phase to the S phase of the cell cycle [36]. This control is crucial for maintaining normal cell growth and division. In healthy cells, p16 regulates the cell cycle by inhibiting the activity of cyclin-dependent kinases 4 and 6 (CDK4/6) [37,38]. When active, these kinases phosphorylate retinoblastoma protein (pRb), a key regulator of the cell cycle. Phosphorylation of pRb leads to its inactivation, allowing the cell to progress from the G1 phase to the S phase and subsequently divide [39,40]. By inhibiting CDK4/6, p16 ensures that a controlled transition occurs, thereby preventing unregulated cell proliferation [41].

In cases of high-risk HPV infection, particularly HPV types 16 and 18, the viral oncoprotein E7binds to pRb, leading to its inactivation [42]. This inactivation mimics the effect of phosphorylation and effectively removes the block that pRb typically places on the cell cycle [43]. The inactivation of pRb by E7 results in the loss of a critical control point in the cell cycle, leading to unregulated cell growth [44]. In response to this disruption, cells increase the production of p16 to counteract uncontrolled progression of the cell cycle [45,46]. Therefore, p16 overexpression is an indirect result of HPV oncogenic activity within the cell [47]. Because p16 overexpression is closely linked to the disruption of cell cycle control by high-risk HPV types, it serves as a surrogate marker of HPV-associated oncogenic activity [48]. In the context of cervical cancer screening, the detection of associated oncogenic activity has been reported [49], and the detection of elevated p16 levels indicates an ongoing HPV infection that has altered the normal regulatory mechanisms of the cell cycle, suggesting the presence of potentially precancerous or cancerous changes in cervical epithelial cells [50].

Ki-67 is a protein that is closely associated with cell proliferation and is widely used as a marker to determine the growth fraction of a cell population [51]. Its utility in the context of cervical cancer and its precursor lesions, such as cervical intraepithelial neoplasia (CIN), is particularly significant. Herein, we present a detailed examination of its role and importance [51]. Ki-67 is present in the nucleus of cells during all active phases of the cell cycle (G1, S, G2, and M phases), although it is absent in cells in the resting phase (G0) [52]. This makes Ki-67 a reliable marker for identifying actively dividing cells. The presence of this protein indicates that the cell is progressing through the cell cycle and is not in a quiescent state. In the cervical epithelium, Ki-67 expression is commonly restricted to the basal and parabasal layers, where cell division normally occurs [53]. However, in dysplastic cells, such as those observed in CIN, Ki-67 expression can be detected in the higher layers of the epithelium. This aberrant expression pattern indicates unregulated cell proliferation, which is a hallmark of precancerous and cancerous changes [54]. The level and pattern of Ki-67 expression are closely correlated with cervical dysplasia grade [55,56]. In low-grade lesions such as CIN1, Ki-67 expression may be slightly increased and remain largely confined to the lower epithelial layers. However, in high-grade lesions, such as CIN2 and CIN3, Ki-67 expression is typically more extensive and is often observed in the higher and more superficial layers of the cervical epithelium. This increase in Ki-67 staining reflects the loss of normal cell cycle control and is a key feature of CIN progression. Assessment of Ki-67 expression is valuable for both the diagnosis and prognosis of CIN and cervical cancer. Elevated Ki-67 levels in cervical cells suggest a higher rate of cell turnover, which is a characteristic of both dysplastic and neoplastic processes. Therefore, Ki-67 staining is often used in conjunction with other histopathological and cytological evaluations to assess the severity of cervical lesions [57,58]. In cervical cancer screening and management, Ki-67 can serve as an adjunct marker to improve the accuracy of cytological diagnosis. It helps differentiate between benign reactive changes and true dysplastic changes, particularly in cases where diagnosis based on morphology alone is challenging [59,60].

Dual staining for p16 and Ki-67 is particularly useful in cervical cancer screening because these two proteins are mutually exclusive in normal cells [56,61,62]. Therefore, their co-expression serves as a specific marker for HPV-mediated oncogenic transformation and indicates a higher risk of cervical cancer [58,63]. This method helps differentiate between transient, harmless HPV infections and persistent infections with a higher potential to progress to high-grade pre-cancer or invasive carcinoma.

### Study Outcomes on Dual Staining Efficacy

Recent advancements in cervical cancer screening have increased the efficacy of the p16/Ki-67 dual-staining technique, particularly for triaging human papillomavirus (HPV)-positive women [64,65]. A growing body of research indicates that this dual-staining method outperforms traditional cytology in several key aspects, notably sensitivity and negative predictive value (NPV), albeit with a trade-off in specificity [14,56]. These findings have significant implications for early detection and management of high-grade cervical lesions.

Dual staining has been demonstrated to have significantly higher sensitivity than cytology for detecting CIN2+ or CIN3+ lesions [23,66]. This higher sensitivity indicates that dual staining is more likely to identify women with high-grade cervical lesions that could progress to cervical cancer if left undetected or untreated.

The most striking advantage of p16/Ki-67 dual staining is its heightened sensitivity for detecting CIN2+ or CIN3+ lesions, which are precursors of cervical cancer [61,64,67]. Studies have consistently indicated that the sensitivity of dual staining is markedly higher than that of traditional cytology. This increased sensitivity ensures a greater likelihood of identifying women with high-grade cervical lesions, which is critical for prompt treatment to prevent the progression to cervical cancer. The ability of dual staining to detect these lesions at an early stage can be attributed to its molecular approach that targets specific cellular changes induced by HPV infection that are occasionally missed during cytological examinations [23,55].

Another significant benefit of the dual-staining technique is its higher NPV for detecting CIN2+ and CIN3+ lesions compared to cytology [56]. This specifies that a negative result from dual staining indicates a low risk of high-grade cervical lesions. In clinical practice, this translates to a reduced likelihood of overlooking significant lesions, thus providing reassurance to both patients and clinicians regarding the absence of serious pathologies [68].

However, the increased sensitivity of p16/Ki-67 dual staining was lower than that of cytology [55,64,69]. This reduced specificity may lead to more false-positive results, where the test indicates the absence of a potential lesion. Consequently, this can result in more women being referred for additional diagnostic procedures such as colposcopy and biopsy, which may not be necessary [70]. Such over-referral can cause undue anxiety in patients and may strain healthcare resources.

Despite concerns regarding specificity, the use of dual staining has been observed to increase the detection of CIN2+ cases [71]. This suggests that dual staining has a superior ability to identify significant cervical lesions compared with cytology, thereby potentially lowering the incidence of cervical cancer through early intervention. The earlier these lesions are detected and appropriately managed, the better the patient outcomes. The p16/Ki-67 dual-staining technique represents a significant step forward in cervical cancer screening, particularly in HPV-positive women [72]. Its superior sensitivity and NPV for detecting high-grade lesions offers the promise of earlier and more accurate identification of women at risk of developing cervical cancer. However, the low specificity of this method necessitates cautious interpretation of the results and underscores the need for balanced clinical decision making. As the screening landscape continues to evolve, the integration of p16/Ki-67 dual staining into existing protocols is expected to refine our approach for the detection and management of precancerous cervical conditions [73].

## 3. Comparative Analysis of Dual Staining versus Traditional Methods

The comparative performance of p16/Ki-67 dual staining and traditional cytology methods in cervical cancer screening has significant differences in sensitivity, specificity, and predictive value (Table 2).

Studies have consistently demonstrated that p16/Ki-67 dual staining exhibits higher sensitivity than traditional cytology for detecting CIN2+ and CIN3+ lesions. For instance, research indicates that the sensitivity of dual staining for detecting CIN2+ can be high at 81.8% compared to the lower sensitivity rates for cytology. This enhanced sensitivity is pivotal for identifying additional cases of significant cervical lesions and potentially reducing the progression of cervical cancer. Although dual staining surpasses cytology in terms of sensitivity, it generally exhibits lower specificity. Dual staining may yield more false-positive results, potentially leading to more women undergoing further diagnostic procedures, such as colposcopy and biopsy, even without high-grade lesions.

The NPV of dual staining was significantly higher than that of traditional cytology, suggesting that a negative result from dual staining is more reliable in assuring a low risk of high-grade cervical lesions [84]. However, its positive predictive value may be lower than that of cytology, indicating a higher likelihood of false positives.

In our comprehensive evaluation, we meticulously compared the sensitivity and specificity of p16/Ki-67 dual staining with those of conventional screening methods, such as cytology and HPV DNA testing [85]. Our analysis reveals that dual staining exhibits superior accuracy in detecting high-grade cervical lesions, attributable to its unique ability to simultaneously identify the overexpression of p16, indicative of transforming HPV infections, and Ki-67, a marker of cellular proliferation [76,86]. This dual biomarker approach enhances the specificity for high-grade lesion detection, thereby reducing the incidence of false positives associated with traditional methods [49]. Our findings underscore the potential of dual staining to significantly improve the efficacy of cervical cancer screening protocols, offering a compelling alternative for triage in HPV-positive individuals.

The incorporation of less specific screening methods, such as HPV testing, has raised concerns regarding potential overdiagnosis and overtreatment in cervical cancer screening programs [87]. Despite the high sensitivity of high-risk HPV testing, it has lower specificity than cytology. This disparity can lead to overdiagnosis, in which HPV-positive women without significant cervical lesions are subjected to additional, possibly unnecessary, diagnostic procedures. Dual staining has emerged as a potential solution to this challenge. Although it has a lower specificity than cytology, its high sensitivity and higher NPV make it a valuable tool for accurately identifying women at a true risk of developing high-grade lesions [88].

This could reduce the number of unnecessary colposcopies and biopsies arising from HPV testing alone [89]. The ideal screening method should strike a balance between high sensitivity (to detect as many true cases as possible) and high specificity (to minimize false positives and unnecessary interventions). Dual staining, with its high sensitivity, is beneficial for initial screening. However, its low specificity necessitates careful interpretation and additional confirmatory testing to avoid overtreatment.

The choice between dual staining and traditional methods depends on the specific context, including patient age, HPV vaccination status, and other risk factors. Clinicians must weigh the benefits of increased sensitivity against the risks associated with lower specificity and tailor their approach to individual patient circumstances.

## 4. Clinical Implications and Future Prospects

### 4.1. Triage Strategies and Follow-Up Results

The integration of p16/Ki-67 dual staining into cervical cancer screening protocols necessitates the re-evaluation of triage strategies, particularly in terms of follow-up and clinical management outcomes [61].

Dual staining has been demonstrated to be effective in refining the triage of HPV-positive women. With its higher sensitivity for detecting CIN2+ lesions, dual staining can identify cases that may be missed by cytology alone. However, this increased sensitivity results in a trade-off in specificity, potentially leading to more biopsies. Comparative studies indicate that the use of dual staining might result in more biopsies being performed because more women are classified as being at risk of developing CIN2+ lesions [56,61,64]. Although this increases the detection rate of significant cervical lesions, it also increases the potential for overtreatment in cases where the lesions might regress spontaneously.

The superior sensitivity of dual staining leads to higher detection rates of CIN2+ lesions, which is crucial for preventing the progression of cervical cancer [90]. The NPV of dual staining is particularly noteworthy, as it ensures a low risk of high-grade lesions in negative cases, thus reducing unnecessary follow-up interventions.

We can propose an algorithm (Figure 1) or sequence of tests that incorporates p16/Ki-67 dual staining within a comprehensive cervical cancer screening strategy. This approach could start with primary HPV testing to identify high-risk HPV infections. Positive cases would then proceed to triage using the p16/Ki-67 dual staining test to detect cellular changes indicative of high-grade lesions. Based on the results, individuals could be directed towards appropriate follow-up actions, ranging from closer surveillance to immediate colposcopy. This approach allows for earlier intervention and more personalized patient management strategies, enhancing the overall effectiveness of cervical cancer prevention programs. This strategy optimizes the detection of clinically relevant lesions while minimizing unnecessary interventions, aligning with current evidence on the effectiveness of combined testing methodologies in the era of HPV vaccination.

### 4.2. Dual Staining in Contemporary Screening Programs

The evolving landscape of cervical cancer screening is influenced by factors such as HPV vaccination and changes in HPV genotype prevalence, and position dual staining is a potentially valuable tool in modern screening strategies [91].

As HPV vaccination programs target specific high-risk HPV types, primarily HPV 16 and 18, changes in the prevalence of other HPV genotypes are expected [92]. The role of dual staining in identifying lesions associated with other high-risk HPV types has become increasingly important. The high sensitivity and NPV of dual staining can help streamline the process of referral to colposcopy clinics. By accurately identifying women at an actual risk of high-grade lesions, dual staining has the potential to reduce the burden on colposcopy clinics by minimizing unnecessary referrals [93]. The incidence of lesions caused by vaccine-targeted HPV will decrease in populations with a high HPV vaccination coverage. Dual staining could play a pivotal role in monitoring and managing lesions caused by non-vaccine HPV types, thereby ensuring the continued effectiveness of screening programs in the post-vaccination era. Ongoing research is essential to elucidate the long-term implications of incorporating dual staining into screening programs [23,71,83,94]. This includes evaluating cost-effectiveness, patient outcomes, and impact on healthcare systems. Additionally, guidelines for the implementation of dual staining need to be developed, considering varying healthcare contexts and population needs.

### 4.3. In a Population with High Proportion of HPV-Vaccinated Individuals

HPV vaccination, particularly against high-risk types like HPV 16 and 18, has significantly reduced the prevalence of these genotypes in vaccinated populations [95]. This shift may lead to a change in the distribution of HPV genotypes responsible for cervical lesions, potentially increasing the relative prevalence of other high-risk HPV types not covered by the vaccine [96]. Understanding these dynamics is crucial for assessing the effectiveness of screening tests like the p16/Ki-67 dual-staining technique, which is designed to detect cellular changes indicative of high-risk HPV infection [23]. The vaccine’s impact on genotype prevalence underscores the need for ongoing surveillance to detect shifts in HPV epidemiology and to evaluate the performance of cervical cancer screening methods in the context of evolving HPV genotype distribution [86].

In the context of HPV vaccination impacting screening strategies, it is essential to adjust cervical cancer screening guidelines to the changing HPV genotype landscape. With the reduction in vaccine-targeted HPV types, there is a need to reassess the sensitivity and specificity of current screening methods, such as the p16/Ki-67 dual-staining test, for detecting high-grade lesions caused by non-vaccine HPV types [97]. Future screening strategies should consider the timing and frequency of screening in vaccinated individuals, potentially incorporating risk-based approaches and utilizing tests that can accurately identify lesions from a broader range of high-risk HPV types. This adjustment will ensure continued effectiveness in cervical cancer prevention in the vaccinated population.

### 4.4. Automated Evaluation in Cytology of Cervical Cancer Screening

Upon reviewing the document on the application of digital pathology in the analysis of p16/Ki-67 dual staining, a deeper exploration reveals that this innovative approach significantly enhances the detection and interpretation of precancerous and cancerous lesions in cervical cytology [98,99]. By leveraging sophisticated image analysis algorithms, digital pathology enables a standardized and automated assessment of stained slides, reducing subjective interpretation variability [100]. This method improves diagnostic accuracy, particularly in identifying high-grade cervical intraepithelial neoplasia. Moreover, the integration of digital pathology facilitates a more efficient workflow, allowing for high-throughput screening and the potential for better resource allocation within healthcare systems. The application of digital pathology in p16/Ki-67 dual staining represents a significant step forward in cervical cancer screening, offering a promising route to improve patient outcomes through early and accurate detection.

## 5. FAM19A4/miR1234-2 Methylation Testing

The hallmark of cervical carcinogenesis is hypermethylation of tumor suppressor genes. Recent clinical investigations have demonstrated that the FAM19A4/miR124-2 methylation test, labeled as CE-In Vitro Diagnostic and standardized, effectively identifies almost all cervical cancers (>98%) and reliably detects advanced CIN lesions [28,101]. Advanced CIN lesions, defined as CIN2/3 lesions exhibiting a cancer-like methylation profile and linked to prolonged HPV infection, may carry a heightened risk of short-term cancer progression. In women aged ≥30 years, this test exhibits a sensitivity of 77% for detecting CIN3. In a prospective clinical cohort study, the absence of FAM19A4/miR124-2 methylation is correlated with a high rate of regression in CIN2/3 lesions [102,103,104]. As a triage strategy, HPV16/18 genotyping is currently recommended, acknowledging the variation in carcinogenic potential among different HPV types. In young women, the FAM19A4/miR124-2 methylation test exhibited a significantly lower positivity rate for CIN2/3 than HPV16/18 genotyping, suggesting its potential to offer higher specificity in reassuring against advanced CIN lesions requiring treatment [103,104]. The ability of the test to reflect the nature of the underlying CIN, specifically distinguishing lesions with high or low short-term cancer progression risk, provides critical guidance for clinical management. With the high specificity observed in methylation-negative CIN2/3 lesions, particularly in young HPV-positive women, the test holds promise for implementing a wait-and-see policy, avoiding unnecessary overtreatment and aligning with the observed high regression rates in this population. Therefore, the FAM19A4/miR124-2 methylation test is a powerful and independent biomarker that provides detailed information for efficient and personalized therapy of cervical lesions, particularly in the context of shifting HPV vaccination guidelines [104].

The VALID-SCREEN study [102] was a retrospective EU multicenter investigation that assessed the clinical utility of the FAM19A4/miR124-2 methylation-based molecular triage test for HPV cervical cancer screening [105]. A previous study conducted on 2384 HPV-positive cervical screening samples from women aged 29–76 years in four EU countries aimed to determine the performance of the test as a substitute or addition to cytology in reflex testing for HPV-positive women. The FAM19A4/miR124-2 methylation test demonstrated a sensitivity of 95% for detecting screening-detected cervical cancers and an overall specificity of 78.3%. The NPV for methylation-negative outcomes in HR-HPV-positive samples was high at 99.9% for cervical cancer, 96.9% for ≥CIN3, and 93.0% for ≥CIN2. With consistent sensitivity for CIN3 across different centers, sample collection media, DNA extraction methods, and HPV screening tests, the FAM19A4/miR124-2 methylation test is an objective and potentially equivalent alternative or supplement to cytology for triaging HPV-positive women in real-life pilot implementation studies.

## 6. Limitations

Importantly, the use of p16/Ki-67 dual staining has implications beyond mere detection; it offers a potential to alleviate the burden on healthcare systems. By enabling more precise triage of patients, particularly those with HPV-positive results, it can reduce unnecessary follow-ups and invasive procedures such as colposcopies and biopsies. This efficiency not only optimizes resource utilization but also spares patients from the anxiety and discomfort associated with these procedures.

However, the journey towards integrating p16/Ki-67 dual staining into routine clinical practice is not without its challenges. The technique, while sensitive, exhibits lower specificity compared to traditional cytology, which raises the possibility of false-positive results. Therefore, its clinical application demands a balanced approach, harmonizing the high sensitivity with the need for specificity. This balance is crucial to ensure patient-centered care that is both effective and economical, avoiding overdiagnosis and overtreatment. The limitations of dual-stained p16/Ki-67 in cervical cancer screening include its potential for false positives in cases of transient HPV infections not progressing to cancer, leading to unnecessary follow-up procedures. Additionally, its effectiveness may vary across different age groups and HPV vaccination statuses, potentially impacting its utility in certain populations. The cost and need for specialized laboratory equipment and expertise could also limit its accessibility, especially in low-resource settings. These factors underscore the importance of considering the context and patient population when integrating dual staining into screening protocols.

## 7. Conclusions

P16/Ki-67 dual staining enhances the detection of precancerous lesions in HPV-positive individuals, improving risk stratification and reducing unnecessary procedures. While it offers increased sensitivity, challenges such as false positives and specificity need balancing to avoid overtreatment. Its integration into clinical practice requires adaptation to diverse patient populations and healthcare settings, considering cost and accessibility. Future efforts should focus on refining its application, supported by evolving screening protocols and advancements in medical technology, to ensure effective, patient-centric cervical cancer screening.

## Figures and Tables

**Figure 1 diagnostics-14-00451-f001:**
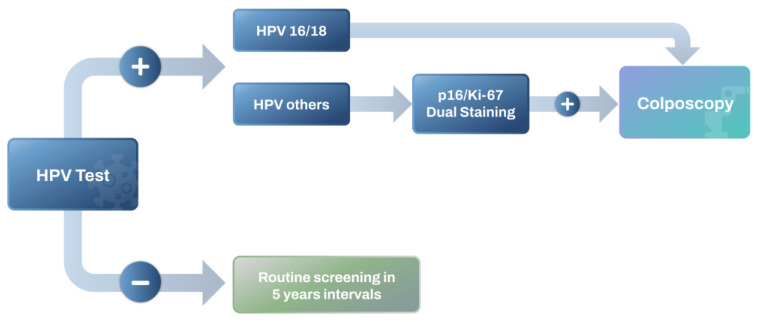
An algorithm for the application of dual-staining cytology in the era of HPV primary screening for cervical cancer screening.

**Table 1 diagnostics-14-00451-t001:** Overview of p116/Ki-67 dual staining in cervical cancer.

Aspect	Details
Technique	Advanced immunocytochemical method for simultaneous detection of p16 and Ki-67 proteins within cervical epithelial cells.
Sensitivity	Higher sensitivity than cytology, particularly effective in detecting CIN2+ or CIN3+ lesions.
Specificity	Lower than cytology, potentially leading to more false-positive results.
Use in triage	Effective in identifying women at risk of developing high-grade cervical lesions, especially in HPV-positive population.
Clinical implication	Facilitates early detection and management of cervical cancer, but requires cautious interpretation because of lower specificity.

**Table 2 diagnostics-14-00451-t002:** Comparative analysis of screening methods [16,21,24,74,75,76,77,78,79,80,81,82,83].

Method	Sensitivity	Specificity	Suitability for Triage
p16/Ki-67 dual staining	Higher	Lower	Highly suitable for HPV-positive women
HPV resting	Higher	Lower, especially in younger populations	Effective, but less specific in younger women
Cytology	Variable, generally lower	Higher	Suitable with limitations in sensitivity

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
