# Peer review of "Enhancing Cervical Cancer Screening: Review of p16/Ki-67 Dual Staining as a Promising Triage Strategy"

_diagnostics, 2024, doi:10.3390/diagnostics14040451_

Round 1
Reviewer 1 Report
Comments and Suggestions for Authors
it is a very interesting topic adressing a real problem
it would be interesting to underline in the discussion sector the relevanance of this test in a population with high proportion of HPV vaccinated individuals
also a algorithm or a sequence of tests including the dual staining could be discussed
Comments on the Quality of English Languageno
Reviewer 2 Report
Comments and Suggestions for Authors
This review is related to very actual topic, triage tests in cervical cancer screening. Although this survey does not bring a lot of new information or something really impactful to this subject. Moreover, some statements are arguable (e.g. comparison dual test with cytology).
1. The main question adressed by the research is how dual staining with p16 and Ki-67 can be used as a triage test. 2. From my point of view the review article does not provide enougn information to fill the gap about cost/effectiveness of dual staining as a triage test because this aspect is the most important to chose among other methods with equal or high sensitivity and/or specificity 3. This article does not bring a lot of new information concerning dual staining in compare with other review articles related to this topic 4. I suppose that the authors should pay more attention to comparison dual staining with other triage tests to provide a conclusion when and who can implement this test as a triage test in the routine practice. I guess that the fundamental aspect with Ki-67 and p16 expression detailed explanation is too flourished in this article. 5. Conclusion should be more consolidated 6. References are appropriate. 7. There are some typos in the tables that should be corrected.
Round 2
Reviewer 2 Report
Comments and Suggestions for Authors
I am satisfied the modification of the article and I suppose that it could be accepted for the publication.
Author Response
Thank you very much for your valuable feedback on our manuscript. We are deeply gratified to hear that you are satisfied with the modifications we have made to the article. Your insightful comments and suggestions have been instrumental in enhancing the quality and clarity of our work.
We are encouraged by your assessment that the article could be accepted for publication. We have strived to address the concerns raised during the review process comprehensively, and it is rewarding to know that our efforts have met your expectations.